# CodeARC: Benchmarking Reasoning Capabilities of LLM Agents for Inductive Program Synthesis

**Anjiang Wei**[1], **Tarun Suresh**[2], **Jiannan Cao**[3], **Naveen Kannan**[1], **Yuheng Wu**[1], **Kai Yan**[2], **Thiago S. F. X. Teixeira**[4], **Ke Wang**[5], **Alex Aiken**[1]
[1]Stanford University    [2]University of Illinois Urbana-Champaign
[3]MIT    [4]Intel    [5]Nanjing University

## Abstract

Inductive program synthesis, or programming by example, requires synthesizing functions from input-output examples that generalize to unseen inputs. While large language model agents have shown promise in programming tasks guided by natural language, their ability to perform inductive program synthesis is underexplored. Existing evaluation protocols rely on static sets of examples and held-out tests, offering no feedback when synthesized functions are incorrect and failing to reflect real-world scenarios such as reverse engineering. We propose CodeARC, the *Code Abstraction and Reasoning Challenge*, a new evaluation framework where agents interact with a hidden target function by querying it with new inputs, synthesizing candidate functions, and iteratively refining their solutions using a differential testing oracle. This interactive setting encourages agents to perform function calls and self-correction based on feedback. We construct the first large-scale benchmark for general-purpose inductive program synthesis, featuring 1114 functions. Among 18 models evaluated, o3-mini performs best with a success rate of 52.7%, highlighting the difficulty of this task. Fine-tuning LLaMA-3.1-8B-Instruct on curated synthesis traces yields up to a 31% relative performance gain. CodeARC provides a more realistic and challenging testbed for evaluating LLM-based program synthesis and inductive reasoning. Our code, data, and models are publicly available at https://github.com/Anjiang-Wei/CodeARC

## 1 Introduction

Inductive reasoning, i.e., the ability to identify patterns and form abstractions from limited examples, is widely recognized as a fundamental aspect of human intelligence (Hayes et al., 2010; Yan et al., 2025). In the context of programming, inductive reasoning underpins the task of synthesizing functions that satisfy given input-output examples and generalize to unseen inputs. This task, commonly referred to as *inductive program synthesis* or programming by example (Manna & Waldinger, 1971; Myers, 1986; Feser et al., 2023; Li & Ellis, 2025), has broad application domains (Wang et al., 2017; Deng et al., 2024).

Recent advances in Large Language Models (LLMs) have led to the emergence of autonomous agents capable of decision-making, multi-step planning, tool use, and iterative self-improvement through interaction and feedback (Chen et al., 2023; Liu et al., 2023e; 2024b; Guo et al., 2024; Xi et al., 2025). While much of the existing work focuses on programming tasks guided by natural language (Chen et al., 2021; Austin et al., 2021; Jimenez et al., 2023; Jain et al., 2024b), we study a fundamentally different problem: inductive program synthesis, where the objective is to infer the target program solely from input-output examples. This setting provides a more rigorous test of inductive reasoning capabilities, as it eliminates natural language descriptions that can trigger retrieval-based completions memorized during model training.

---

Correspondence to: anjiang@cs.stanford.edu

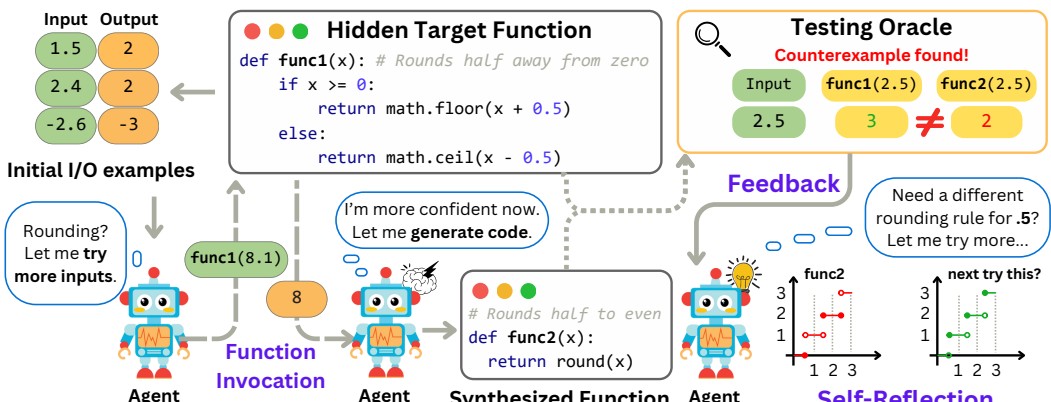

Figure 1: **Overview of CodeARC.** Our framework evaluates LLMs' reasoning capabilities in inductive program synthesis. The agent begins with input-output examples, interacts with a hidden target function via function calls, and uses a differential testing oracle to check the correctness of the synthesized function for self-reflection and refinement.

Designing an effective evaluation protocol for inductive program synthesis with LLMs is inherently challenging, as multiple valid functions may satisfy a given set of input-output examples (as demonstrated in Section 5.2). The state-of-the-art protocol (Li & Ellis, 2025), which evaluates synthesized functions on held-out test cases after presenting 10 fixed input-output examples, has several notable limitations. First, a small, static set of input-output examples may underspecify the target functions, especially for those with complex logic. Moreover, held-out tests may fail to reveal subtle semantic discrepancies between the generated and intended implementations. In addition, when the model produces an incorrect solution, it receives no feedback and has no opportunity to revise or explore alternatives. Finally, existing benchmarks for inductive program synthesis (Gulwani, 2011; Wang et al., 2017; Wong et al., 2021; Deng et al., 2024; Li & Ellis, 2025) are focused on domain-specific tasks and do not assess the ability of LLMs to synthesize functions written in general-purpose programming languages.

To address the limitations of existing evaluation protocols for inductive program synthesis, we introduce CodeARC, the *Code Abstraction and Reasoning Challenge,* inspired by real-world scenarios such as decompilation and reverse engineering (Lin et al., 2010; Mantovani et al., 2022; Al-Kaswan et al., 2023). In such settings, an agent is given a binary executable (without source code) and must synthesize equivalent source code by observing input-output behavior. Instead of relying on a fixed dataset, the agent can query the binary with new inputs, invoke a differential testing oracle, and use counterexamples for iterative refinement. This setup parallels the classic learnability framework of queries and counterexamples (Angluin, 1987; De la Higuera, 2010), here applied to program synthesis.

Figure 1 illustrates how CodeARC instantiates this process: LLM-based agents begin with an initial set of input-output examples, query the ground-truth function for more examples, and debug synthesized code using a differential testing oracle. We impose fixed budgets on both the number of observable input-output examples and the number of testing oracle invocations for self-debugging. The task requires agents to proactively generate inputs (function calls) and revise solutions based on feedback (self-reflection). This interactive setup offers a more realistic alternative to prior static evaluation protocols.

We construct the first comprehensive dataset for general-purpose inductive program synthesis, featuring 1114 functions with initial input-output examples. Our benchmark targets general programming tasks and employs two state-of-the-art differential testing tools (Lukasczyk & Fraser, 2022; Etemadi et al., 2024) for correctness evaluation.

Our experiments demonstrate that CodeARC poses a significant challenge for LLM-based inductive program synthesis. Among the 18 models evaluated, OpenAI o3-mini performs the best overall, yet only achieves 52.7% success rate. We further conduct ablation studies to

analyze how budgets on the number of input-output examples and oracle calls affect model performance. To enhance model capabilities, we generate synthetic fine-tuning data with curated synthesis traces that capture the reasoning steps. We show that supervised fine-tuning on LLaMA-3.1-8B-Instruct yields up to a 31% relative performance improvement.

In summary, our contributions are as follows:

- **Interactive evaluation protocol for inductive program synthesis.** We introduce a setup where agents start with fixed input-output examples but can generate new inputs to query ground-truth functions and invoke a differential testing oracle to self-correct their solutions. This setup brings the task closer to a real-world setting, e.g. reverse-engineering.
- **General-purpose benchmark with extensive evaluation.** We construct the first large-scale, general-purpose benchmark for this task, including 1114 diverse functions. Among the 18 models evaluated, o3-mini achieves the best overall performance but still only reaches a success rate of 52.7%.
- **Synthetic data and fine-tuning.** To boost model performance, we generate synthetic fine-tuning data containing curated synthesis traces that capture both function invocations and reasoning steps. We show that fine-tuning on LLaMA-3.1-8B-Instruct yields up to a 31% relative performance improvement.

## 2 Related Work

**Inductive Program Synthesis**    Traditional inductive program synthesis methods rely solely on input-output examples, without natural language input. They focus on domain-specific tasks like string and data transformations (Gulwani, 2011; Singh & Gulwani, 2016; Yaghmazadeh et al., 2016), SQL (Wang et al., 2017), visual programming (Wang et al., 2019), and quantum computing (Deng et al., 2024). These approaches typically define a domain-specific language and use tailored search algorithms to prune the space efficiently (Feser et al., 2015; Polikarpova et al., 2016; Feng et al., 2018; Guria et al., 2023; Mell et al., 2024). In contrast, we introduce the first general-purpose program synthesis benchmark for LLM-powered agents.

**LLM Benchmarks for Code**    Most LLM benchmarks, such as HumanEval+ (Chen et al., 2021; Liu et al., 2023b), MBPP+ (Austin et al., 2021; Liu et al., 2023b), APPS (Hendrycks et al., 2021), and others (Li et al., 2022; Liu et al., 2023c; Li et al., 2023; Jain et al., 2024a; Banerjee et al., 2025; Ni et al., 2024b; Liu et al., 2023d; Du et al., 2023; Suresh et al., 2025b; Zhuo et al., 2024; Yin et al., 2022; Lai et al., 2023; Patil et al., 2025; Ugare et al., 2024; 2025; Wei et al., 2025d; Wu et al., 2025), evaluate code generation from natural language. Beyond generation, tasks like I/O prediction (Gu et al., 2024; Liu et al., 2024a), execution prediction (Liu et al., 2023a; La Malfa et al., 2024; Ni et al., 2024a; Ding et al., 2024), bug localization (Suresh et al., 2025a), and program equivalence (Wei et al., 2025a) have also been studied. In contrast, we focus on predicting function bodies purely from input-output examples, without natural language. Prior work (Li & Ellis, 2025; Barke et al., 2024) targets domain-specific tasks, while we introduce a general-purpose benchmark with an interactive evaluation protocol.

**LLM Benchmarks for Reasoning**    LLMs are widely benchmarked on reasoning tasks across domains, including commonsense (Talmor et al., 2018), mathematical (Cobbe et al., 2021), and logical (Han et al., 2022; Miao et al., 2021; Liu et al., 2020; Parmar et al., 2024; Wei et al., 2025e). Inductive reasoning, a core cognitive skill that generalizes from limited examples (Hayes et al., 2010), is increasingly studied in LLMs (Li et al., 2024; Ma et al., 2024; Xiao et al., 2024; Cai et al., 2024; Shao et al., 2024). ARC (Chollet, 2019) is a prominent benchmark for abstract pattern induction. Our work shares this goal but is for inductive program synthesis.

**LLM-powered Agents**    LLM-based agents have shown strong performance in domains like web navigation (Zhou et al., 2024a;b), code generation (Zhang et al., 2023; Jimenez et al., 2024), performance optimization (Wei et al., 2025b;c), and ML experimentation (Huang et al., 2024). They interact with environments, invoke functions, make decisions, and self-reflect (Madaan et al., 2023; Paul et al., 2023; Xie et al., 2023; Huang et al., 2022; Shinn et al.,

2023). We introduce the first benchmark to systematically evaluate agents' capabilities in inductive program synthesis, providing a rigorous testbed for inductive reasoning and program synthesis.

# 3 Method

## 3.1 Problem Definition of Inductive Program Synthesis

We formalize the inductive program synthesis task as follows. Let $f^*$ be a *hidden* ground-truth function that maps inputs $x \in \mathcal{X}$ to outputs $y \in \mathcal{Y}$. The synthesizer is given an initial set of input-output examples $\mathcal{E}_0 = \{(x_i, y_i)\}_{i=1}^n$, where $y_i = f^*(x_i)$, and the goal is to synthesize a program $\hat{f}$ such that $\hat{f} \equiv f^*$, i.e.,

$$\forall x \in \mathcal{X}, \quad \hat{f}(x) = f^*(x).$$

To evaluate whether a synthesized function $\hat{f}$ is correct, we introduce a *differential testing oracle* $\mathcal{O}$. The oracle takes as input both the synthesized function $\hat{f}$ and the hidden ground-truth function $f^*$ and attempts to identify inputs on which their behaviors differ. Formally, the oracle operates as follows:

$$\mathcal{O}(f^*, \hat{f}) = \begin{cases} \text{Pass,} & \text{if } \forall x \in \mathcal{X}_{\text{test}}, \ \hat{f}(x) = f^*(x); \\ \text{Fail}(x), & \text{if } \exists x \in \mathcal{X}_{\text{test}} \text{ such that } \hat{f}(x) \neq f^*(x), \end{cases}$$

where $\mathcal{X}_{\text{test}} \subseteq \mathcal{X}$ is a set of test inputs dynamically selected by the oracle.

Unlike fixed held-out test sets used in prior work, the differential testing oracle conditions on both $f^*$ and the candidate $\hat{f}$, generating targeted inputs to expose discrepancies. On failure, it returns a counterexample $x \in \mathcal{X}_{\text{test}}$ such that $\hat{f}(x) \neq f^*(x)$. Note that program equivalence checking is fundamentally undecidable, and thus no perfect oracle exists. To approximate oracle functionality, we adopt two state-of-the-art differential testing tools, enabling a more robust and practical evaluation than prior work or reliance on a single tool (see Appendix A.3).

## 3.2 Interactive Evaluation Protocol for LLM Agents

To evaluate the capabilities of LLM-based agents in inductive program synthesis, we introduce an interactive protocol. This protocol extends beyond static evaluation settings by enabling dynamic interaction with the hidden ground-truth function and the differential testing oracle.

**Initial Information.** At the beginning of the task, the agent is provided with an initial set of input-output examples $\mathcal{E}_0 = \{(x_i, y_i)\}_{i=1}^n$, as explained previously. This set serves as partial information about the target function.

**Action Space.** During evaluation, the agent may take two types of actions. First, it may query the ground-truth function $f^*$ at a chosen input $x \in \mathcal{X}$, and obtain the corresponding output $f^*(x)$, thereby augmenting its observed set of input-output pairs. Second, it may synthesize a candidate program $\hat{f}$ and invoke the differential testing oracle $\mathcal{O}(f^*, \hat{f})$, which returns PASS if no discrepancies are found on a dynamically generated test set, or FAIL with a counterexample $x \in \mathcal{X}_{\text{test}}$ such that $\hat{f}(x) \neq f^*(x)$.

**Self-Reflection.** If the oracle returns FAIL, a counterexample $\delta$, which is a tuple of $(x, \hat{f}(x), f^*(x))$ will be provided to the agent. This counterexample helps the agent to self-reflect and revise its current hypothesis, either by issuing additional queries $f^*$ or synthesizing new programs. The ability to take such feedback is crucial for iterative refinement.

**Budget Constraints.**   The agent operates under two budget parameters: $B_{\text{io}}$ and $B_{\text{oracle}}$. $B_{\text{io}}$ limits the total number of input-output examples that the agent can observe from the ground-truth function $f^*$, while $B_{\text{oracle}}$ limits the number of invocations to the differential testing oracle $\mathcal{O}$.

**Evaluation Metrics.**   The task is considered successful if the final synthesized program $\hat{f}$, produced within the given budgets, receives a PASS from the differential testing oracle $\mathcal{O}(f^*, \hat{f})$. We assess LLM agent performance along two dimensions: correctness and efficiency, prioritizing correctness. Correctness is the success rate, i.e., the proportion of problems solved within the budget. Efficiency is the average number of input-output queries and oracle invocations per problem.

### 3.3 Benchmark Preparation

Our benchmark is designed to evaluate LLM agents on the inductive synthesis of general-purpose Python programs. This contrasts with prior benchmarks that focus on domain-specific tasks or programs written in domain-specific languages, such as string manipulation and SQL query generation (Yaghmazadeh et al., 2017; Deng et al., 2024; Li & Ellis, 2025).

**Programs for Synthesis.**   We curate a diverse collection of Python functions sampled from three established code generation benchmarks: HumanEval (Chen et al., 2021), MBPP (Austin et al., 2021), and APPS (Hendrycks et al., 2021). HumanEval and MBPP primarily consist of simple, entry-level programming tasks, whereas APPS contains more challenging problems that resemble competition-level code exercises. Importantly, we extract only the function bodies from these benchmarks and do not use any accompanying natural language descriptions.

**Annotated vs. Anonymized.**   To assess the extent to which function names help the LLM agent synthesize the correct program, we construct two versions of the benchmark. In the *annotated* version, function names that reflect the intended functionality of the task (e.g., `is_palindrome`) are made available to the agent. In the *anonymized* version, all function names are replaced with a generic identifier (i.e., `solution`). This design isolates the influence of identifier cues on synthesis performance. We report results on both versions.

**Initial Input-Output Examples.**   Each synthesis task includes 10 fixed input-output examples that specify the target function's expected behavior. We use GPT-4o to generate diverse inputs and execute the original function to obtain the corresponding ground-truth outputs.

**Synthetic Data Generation.**   We generate a synthetic dataset for fine-tuning (described in more detail in Section 3.4) by first collecting 50 seed Python functions that are *disjoint* from the evaluation set. Using these seeds, we prompt GPT-4o to synthesize a diverse set of new functions, yielding 10,000 candidates. For each generated function, we additionally instruct GPT-4o to produce 10 representative inputs that expose the function's behavior and highlight patterns in its input-output relationships. These inputs are executed to verify the executability of the functions, and we discard any that fail at this stage. To reduce redundancy, we deduplicate the dataset based on function names, finally resulting in 5,405 unique Python functions used for fine-tuning.

### 3.4 Fine-Tuning on Synthetic Data

We evaluate whether fine-tuning on curated synthesis traces, which capture both function calls and reasoning, improves LLM performance on CodeARC, using a distillation approach that imitates the reasoning of a teacher model with access to $f^*$, the ground-truth function.

During training, we first run the interactive evaluation protocol described in Section 3.2 with a frozen teacher model. Unlike a standard evaluation, we prepend a set of task-specific instructions $P_{f^*}$ to each teacher prompt. This prefix includes the function body of $\hat{f}^*$ and explicitly instructs the teacher to (1) query $f^*$ on informative inputs, (2) explain the rationale

behind those queries, and (3) synthesize the full implementation of $f^*$ only when confident that the correct logic can be inferred from the accumulated input-output pairs. The student model, which is the only model we fine-tune, learns to mimic the teacher's reasoning and synthesis behavior without seeing the teacher's prompt that includes the target function.

We provide the teacher with access to $f^*$ because we find that, in many cases, the model struggles to solve the task independently. Without knowledge of the ground truth, the teacher is often too weak to generate meaningful queries or explanations, limiting the effectiveness of the resulting supervision.

While executing the evaluation protocol, we record the multi-turn conversation history $C_T$, which comprises the teacher model's prompts and responses. Let $n$ be the total number of turns in $C_T$ and denote by $x^i$ the sequence of tokens in the $i$th turn. Furthermore, let $p$ represent the number of tokens in the teacher-specific instruction prefix $P_{f^*}$.

We fine-tune the student model using a language modeling objective by minimizing the negative log-likelihood of predicting the next token in $C_T$:

$$\mathcal{L} = -\sum_{i=1}^{n} \sum_{j=p+1}^{|x^i|} \log P\left(x_j^i \mid C_{T,<i}, x_{<j}^i\right).$$

Here, $P\left(x_j^i \mid C_{T,<i}, x_{<j}^i\right)$ denotes the probability of generating token $x_j^i$ given all tokens from previous turns, $C_{T,<i}$, and the tokens preceding $x_j^i$ in the current turn, $x_{<j}^i$. By starting the inner sum at $j = p + 1$, the teacher-specific instructions $P_{f^*}$ are excluded from the loss computation, ensuring that the training signal comes only from the parts of $C_T$ available during inference.

## 4 Experiment Setup

We construct two versions (annotated and anonymized) of the 1114 Python functions, drawn from HumanEval+, MBPP+ (Liu et al., 2023b), and APPS (Hendrycks et al., 2021). Table 1 summarizes key statistics. Unlike prior work in program synthesis that often focuses on domain specific languages and constrained settings, our benchmark consists of programs written in Python, a general-purpose language that captures a broader range of real-world algorithms and tasks.

| Source | Functions | Lines of Code | | |
|--------|-----------|-----|-----|-----|
| | | Min | Max | Avg |
| HumanEval+ | 78 | 7 | 56 | 18.5 |
| MBPP+ | 131 | 2 | 21 | 3.9 |
| APPS | 905 | 2 | 74 | 9.5 |
| Annotated | 1114 | 2 | 74 | 9.5 |
| Anonymized | 1114 | 2 | 74 | 9.5 |

Table 1: Number of functions and lines of code statistics for each benchmark source across both dataset versions.

For the main evaluation (Section 5.1), we provide 10 initial input-output examples and set the query budget to 30 input-output pairs and 2 oracle calls ($B_{io} = 30$, $B_{oracle} = 2$), chosen based on practical constraints such as API cost and runtime. Section 5.3 reports ablation studies on both budgets. We use two state-of-the-art differential testing tools, PYNGUIN (Lukasczyk & Fraser, 2022) and MOKAV (Etemadi et al., 2024). For supervised fine-tuning on synthetic reasoning trajectories, we use gpt-4o as the teacher model and LLaMA-3.1-8B-Instruct as the student. See Appendix A for further details, including prompts, fine-tuning parameters, and additional results on the differential testing tools.

## 5 Results

### 5.1 Main Results

Table 2 shows the results for 18 large language models on CodeARC. The order is sorted based on the success rate on the anonymized dataset. We also report the average number of observed input-output examples and oracle invocations. Our findings are as follows:

| Model | Annotated Dataset | | | Anonymized Dataset | | |
|---|---|---|---|---|---|---|
| | #I/O | # Oracle | Success (%) | #I/O | # Oracle | Success (%) |
| Llama-3.2-3B | 28.3 | 1.9 | 11.0 | 29.3 | 2.0 | 4.8 |
| Mixtral-8x7B | 27.4 | 1.9 | 20.3 | 28.5 | 1.9 | 12.0 |
| Llama-3.1-8B | 28.0 | 1.8 | 19.3 | 28.6 | 1.9 | 13.7 |
| Mixtral-8x22B | 26.7 | 1.8 | 25.1 | 28.1 | 1.9 | 15.0 |
| QwQ-32B | 24.6 | 1.8 | 20.0 | 25.7 | 1.9 | 15.4 |
| Qwen2.5-7B | 26.9 | 1.8 | 29.2 | 28.3 | 1.9 | 15.8 |
| Llama-3.2-11B | 27.3 | 1.8 | 24.9 | 28.3 | 1.9 | 16.1 |
| gpt-4o-mini | 27.0 | 1.8 | 26.1 | 27.9 | 1.8 | 18.5 |
| Llama-3.2-90B | 26.2 | 1.8 | 28.4 | 27.7 | 1.9 | 19.7 |
| Llama-3.1-70B | 26.9 | 1.8 | 30.1 | 27.9 | 1.9 | 20.0 |
| Qwen2.5-72B | 25.5 | 1.7 | 30.1 | 27.1 | 1.8 | 21.6 |
| Llama-3.1-405B | 24.2 | 1.7 | 38.6 | 26.0 | 1.8 | 26.7 |
| gpt-4o | 23.4 | 1.7 | 37.8 | 25.2 | 1.8 | 28.7 |
| DeepSeek-V3 | 23.7 | 1.7 | 37.7 | 25.1 | 1.8 | 29.5 |
| claude3.7-sonnet | 23.6 | 1.7 | 39.0 | 24.6 | 1.7 | 33.8 |
| DeepSeek-R1 | 18.6 | 1.6 | 49.8 | 20.3 | 1.7 | 41.3 |
| o1-mini | 21.0 | 1.6 | 53.2 | 21.5 | 1.6 | 47.7 |
| o3-mini | 15.6 | 1.5 | 59.5 | 16.0 | 1.6 | 52.7 |

Table 2: Success rates of LLMs on CodeARC using both annotated and anonymized datasets. We also report the average number of observed input-output examples and oracle invocations. All open-source models are instruction-tuned.

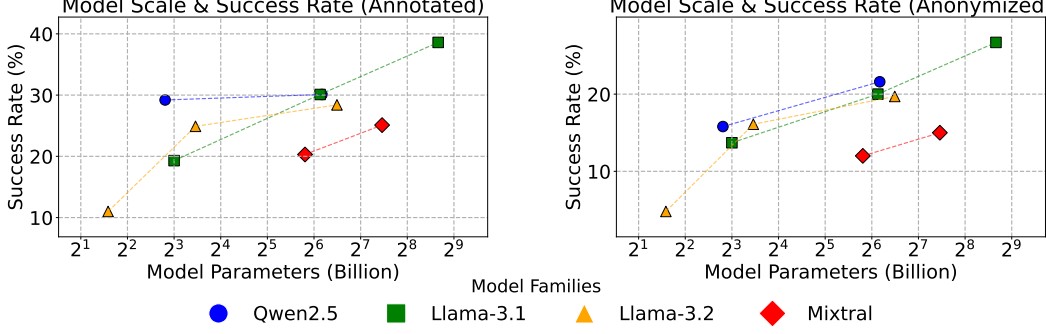

Figure 2: Scaling trend on CodeARC.

**Reasoning models perform best.** Reasoning models (o3-mini, o1-mini, DeepSeek-R1) achieve the highest success rates, all exceeding 40% on the anonymized dataset. They also require fewer I/O examples and oracle calls, indicating greater accuracy and efficiency.

**CodeARC is a challenging benchmark.** Among the 18 evaluated models, only OpenAI's o3-mini achieves over 50% success on both datasets (i.e., 59.5% on the annotated and 52.7% on the anonymized) while all other models fall short of this threshold. This underscores the difficulty of the task and reveals the limitations of current models in inductive reasoning.

**Anonymization of function names reduces performance, but trends persist.** All models show a modest drop in success rate on the anonymized dataset. However, the overall ranking remains largely consistent. This suggests that while the presence of meaningful function names provides some benefit, strong inductive reasoning remains the main factor behind high performance on this synthesis benchmark.

**Scaling up model size improves performance.** Larger models generally achieve better performance, as shown in Figure 2 (log-scale x-axis). All model families exhibit scaling trends, though with varying consistency. Llama-3.1 scales steadily, while Llama-3.2 plateaus at larger sizes, likely due to its multimodal focus. Qwen2.5 shows clearer scaling on

| Metric | # Problems (%) |
|---|---|
| **Pass1:** Initial I/O Examples | 1506 (67.6%) |
| **Pass2:** Testing Oracle | 866 (38.9%) |
| $\Delta =$ **Pass1** $-$ **Pass2** | **640 (28.7%)** |

Table 3: Number of problems (in both datasets) where the synthesized function passes the initial examples compared to the oracle.

| Model | Success Rate (%) | | |
|---|---|---|---|
| | 10 I/O | 20 I/O | 30 I/O |
| o1-mini | 43.7 | 49.6 | 50.5 |
| o3-mini | 51.3 | 53.8 | 56.1 |

Table 4: Success rates (%) with varying budgets on the observable input-output examples (on both datasets).

anonymized data, where reasoning is required over memorization, highlighting model size's impact on generalization.

## 5.2 Do Initial Input-Output Examples Underspecify the Target Function?

To assess whether 10 input-output examples (Li & Ellis, 2025) suffice to specify the target function, we evaluate the first synthesized function of o3-mini, i.e., the strongest model in our study. Functions that pass the initial examples but fail under oracle testing indicate under-specification, motivating the need for additional examples or oracle-guided feedback. As shown in Table 3, 67.6% pass the initial 10 input-output example tests, but only 38.9% pass the oracle, revealing 640 cases (28.7%) where the initial examples fail to uniquely specify the target function. These findings show that initial input-output examples often under-specify program behavior, motivating additional queries and oracle-guided feedback for reliable evaluation. This motivates the design of our interactive evaluation protocol.

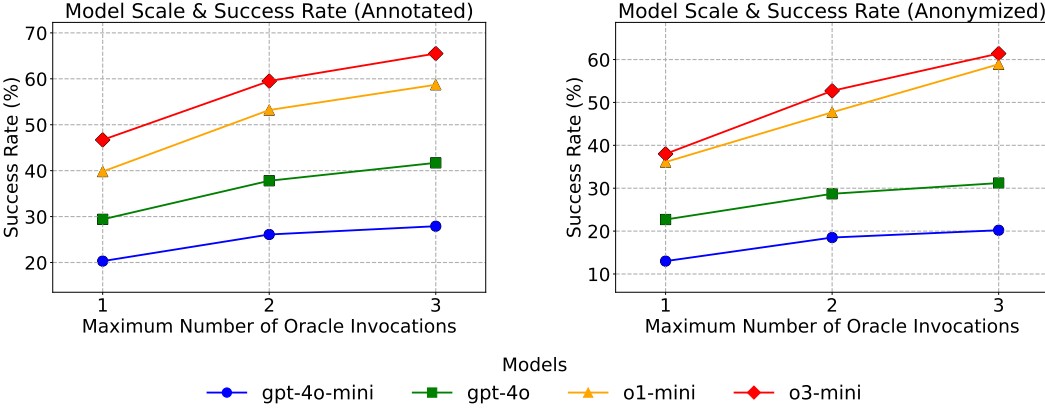

Figure 3: Success rates (%) of LLM models across varying numbers of oracle invocations.

## 5.3 Ablation Study: Input-Output Queries and Oracle Feedback

We perform two ablation studies to evaluate how varying the budgets for querying ground-truth functions and invoking the oracle impacts performance.

**Effect of Input-Output Query Budget.** We evaluate o3-mini and o1-mini with input-output budgets of 10, 20, and 30, using the same setup as Section 5.1. Table 4 shows that success rates improve consistently with more examples.

**Effect of Oracle Invocation Budget.** We vary the number of allowed oracle invocations and report success rates in Figure 3 for four models on both datasets. More oracle calls consistently improve performance, showing that counterexamples from differential testing are valuable for guiding iterative refinement.

These results demonstrate that incorporating both querying mechanisms and oracle feedback consistently enhances overall performance. This improvement underscores the importance

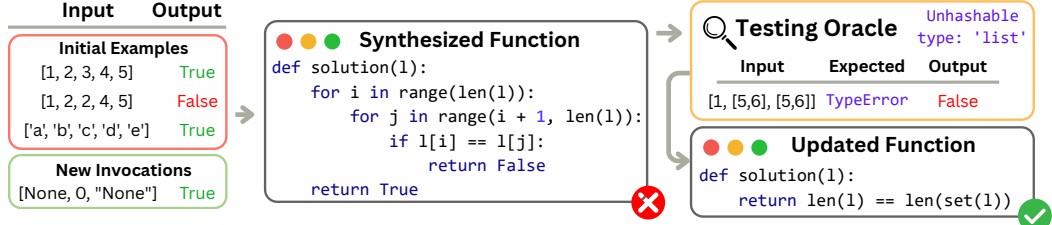

Figure 4: **Case Study.** The model queries edge cases, synthesizes a comparison function, receives a counterexample from the oracle, and corrects it with a set-based solution.

of adopting an interactive evaluation protocol rather than relying solely on static, one-shot evaluation approaches.

## 5.4 Performance of Fine-Tuned Models

Table 5 shows that fine-tuning the LLaMA-3.1-8B-Instruct model on curated synthesis traces yields consistent improvements across both datasets. The larger gain on the annotated variant suggests that fine-tuning is particularly effective when semantically informative identifiers are present. Notably, this performance gap emerges despite both datasets being evaluated under the same model architecture and train-

| Dataset | Success Rate (%) | | |
| --- | --- | --- | --- |
| | Base Model | Fine-Tuned | Rel. Δ |
| Annotated | 19.3 | 25.3 | +31% |
| Anonymized | 13.7 | 15.0 | +9.5% |

Table 5: Success rates of LLaMA-3.1-8B-Instruct and its fine-tuned variant on annotated and anonymized datasets. Fine-tuning improves performance, especially on the annotated dataset.

ing methodology discussed in Section 3.4. These results indicate that while fine-tuning helps, there remains substantial room for further improvement. This suggests that future research may focus on enhancing the quality and diversity of the fine-tuning dataset, particularly for the anonymized variant, where gains are more limited. Another promising direction is to explore reinforcement learning approaches (Wei et al., 2025c) that optimize for higher synthesis success rates, potentially overcoming limitations of supervised fine-tuning alone.

To assess whether fine-tuning affects general code generation capabilities, we additionally evaluate the models on BigCodeBench (Zhuo et al., 2024). Unlike our inductive program synthesis benchmark, BigCodeBench measures traditional code generation from natural language descriptions, covering diverse function calls and library usage. Our synthetic fine-tuning dataset is constructed independently, with no overlap with BigCodeBench, providing a clean test of the fine-tuned model's performance on other coding tasks.

On BigCodeBench, the base model achieves a pass@1 score of 40.1%, while the fine-tuned model attains 39.6%, a marginal decrease of 0.5 percentage points. This aligns with prior findings on catastrophic forgetting (Kotha et al., 2024), a well-documented phenomenon in which training on a new task can impair performance on previously learned ones. We consider this small degradation acceptable given the gains in inductive program synthesis.

## 5.5 Case Study

Figure 4 shows an interaction trace from our benchmark. The model starts by querying the ground-truth function with edge-case inputs, aiming to probe its behavior beyond the initial examples. It then synthesizes a candidate solution using pairwise comparisons, which passes the given examples but fails on a counterexample with unhashable elements. From the error message, the model correctly infers that the ground-truth function raises a TypeError, while its own does not. On the second attempt, it reasons that using a set simplifies the uniqueness check and synthesizes the correct set-based function. This case illustrates how the model combines function invocation and oracle feedback to perform inductive program synthesis. The full trace is in Appendix A.1.

## 6 Discussion

### 6.1 Results on BigCodeBench for Broader Domain Coverage

Our interactive protocol is domain-agnostic and readily extensible to any Python program. To further increase the domain coverage, we extend the original CodeARC dataset with 200 Python functions randomly sampled from BigCodeBench (Zhuo et al., 2024), which includes problems involving scientific computing (e.g., NumPy, SciPy, pandas), machine learning libraries (e.g., sklearn), visualization (e.g., matplotlib), and system libraries. This addition enhances domain diversity beyond traditional algorithm-focused problems.

Table 6 shows that reasoning models (o3-mini, o1-mini) achieve lower success rates on BigCodeBench compared to their performance on the original CodeARC benchmarks. In contrast, non-reasoning models perform better on BigCodeBench. We attribute this to the nature of BigCodeBench tasks, which often involve domain-specific APIs and library usage, where non-reasoning models perform better. Notably, our fine-tuned model shows a substantial gain over the base model on BigCodeBench, indicating that the model fine-tuned on synthesis reasoning traces can generalize to unseen datasets.

| Model | CodeARC (Original) | BigCodeBench |
|---|---|---|
| LLaMA-3.1-8B Base | 19.3 | 32.5 |
| Fine-tuned LLaMA | 25.3 | 40.0 |
| gpt-4o | 37.8 | 46.5 |
| o1-mini | 53.2 | 40.0 |
| o3-mini | 59.5 | 43.0 |

Table 6: Success rates (%) on the original CodeARC benchmark and BigCodeBench, which covers diverse domains such as scientific computing and machine learning, extending beyond algorithm-focused problems.

### 6.2 Impact of Providing Access to a Python Interpreter

Our default evaluation restricts access to external tools such as a Python interpreter, though the CodeARC protocol can easily incorporate them via additional actions. To assess the impact, we added a code execution action, updated the prompts, and evaluated 100 sampled problems from the dataset.

Results shown in Table 7 suggest that providing access to a Python interpreter does not uniformly improve model performance. For instance, while o3-mini benefits modestly (+2.0%), gpt-4o's performance slightly decreases (-1.5%). This outcome highlights that expanding the agent's action space may sometimes lead to suboptimal behavior.

| Model | w/o Interpreter | w/ Interpreter |
|---|---|---|
| gpt-4o | 48.5 | 46.0 |
| o3-mini | 69.0 | 71.0 |

Table 7: Success rates (%) with and without access to a Python interpreter.

We note that internal code simulation is a natural part of the inductive program synthesis process. Our benchmark is designed to evaluate inductive reasoning capability, and we believe that requiring models to reason without relying on external execution tools remains a meaningful and challenging setting.

## 7 Conclusion

We introduce CodeARC, a new framework for evaluating LLMs on inductive program synthesis through interactive input generation and self-correction. Unlike static protocols, CodeARC allows agents to query a ground truth function and use a differential testing oracle to get feedback for iterative refinement. Designed to assess inductive reasoning from input-output examples, our benchmark covers 1114 diverse and general-purpose functions and evaluates 18 language models. The best-performing model, OpenAI o3-mini, achieves a success rate of 52.7%. Fine-tuning LLaMA-3.1-8B-Instruct on curated synthesis traces results in a 31% relative performance gain. CodeARC provides a more realistic and challenging testbed for evaluating LLM-based inductive program synthesis.

## Acknowledgments

We thank Yuhui Zhang and Xiaohan Wang for the discussion. This work was partially supported by a Google Research Award.

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

# A Appendix

## A.1 Case Study

In the below example, the model first invokes the ground-truth function with 10 more inputs, then tries to synthesize the function, and then gets back the counter-example from the differential testing oracle. Its second trial then passes the oracle's test.

**Prompt:**

You are a software engineer tasked with implementing a blackbox Python function `solution`. Your goal is to implement this function correctly, ensuring it works for all possible valid inputs, not just the example inputs provided.

Your primary goal is to get the function implementation exactly right. Your secondary goal is to minimize the number of function invocations and implementation attempts.

**GIVEN EXAMPLES:**

```python
print('Result 1: ' + str(solution([1, 2, 3, 4, 5])))
print('Result 2: ' + str(solution([1, 2, 2, 4, 5])))
print('Result 3: ' + str(solution(['a', 'b', 'c', 'd', 'e'])))
print('Result 4: ' + str(solution(['apple', 'banana', 'apple'])))
print('Result 5: ' + str(solution([])))
print('Result 6: ' + str(solution([10, 20, 30, 40, 50, 60])))
print('Result 7: ' + str(solution([10, 20, 30, 30, 50, 60])))
print('Result 8: ' + str(solution(['x', 'y', 'z'])))
print('Result 9: ' + str(solution([1, 1, 1, 1])))
print('Result 10: ' + str(solution(list(range(100)))))
```

Corresponding outputs:

```
Result 1: True
Result 2: False
Result 3: True
Result 4: False
Result 5: True
Result 6: True
Result 7: False
Result 8: True
Result 9: False
Result 10: True
```

**Strategy Overview:**
You should work in two phases:
1. Exploration Phase: use function invocations strategically to fully understand the function's behavior.
2. Implementation Phase: only after a thorough understanding, attempt an implementation.

This is the first iteration. You must first reason and then decide between one of the following two options:
1. Generate more function invocations in the same format as the examples. You can generate up to 20 additional function invocations across all iterations.
2. Generate an implementation of `solution`. You have only 1 debugging check total. If your implementation is incorrect, you lose your check and are shown a counterexample. If you attempt an implementation and have no debugging checks left and your implementation is incorrect, you cannot take any more actions. This attempt will be considered your final answer.

**Assistant:**

I'll start with the Exploration Phase and generate additional strategic invocations to better understand the function's behavior over a wide range of inputs. I want to explore cases like unique negative and non-negative numbers, duplicate negative numbers, duplicate empty strings, duplicate boolean values, and unique special character strings. I also want to verify that single element lists should be unique.

**INVOCATIONS:**

```python
print('Result 11: ' + str(solution([-1, -2, 0, 1])))
print('Result 12: ' + str(solution([-1, -2, -1])))
print('Result 13: ' + str(solution([1, 1.0, 2])))
print('Result 14: ' + str(solution([(1,), (2,), (1,)])))
print('Result 15: ' + str(solution([42])))
print('Result 16: ' + str(solution(["", "a", "b"])))
print('Result 17: ' + str(solution(["", "", "b"])))
print('Result 18: ' + str(solution([True, False, False])))
print('Result 19: ' + str(solution(['!', '@', '#', '$'])))
print('Result 20: ' + str(solution([None, 0, "None"])))
```

**Prompt:**

Here are the outputs of the function invocations from the previous iteration:

```
Result 11: True
Result 12: False
Result 13: False
Result 14: False
Result 15: True
Result 16: True
Result 17: False
Result 18: False
Result 19: True
Result 20: True
```

STRATEGY UPDATE: Based on the new data above, you should update your understanding of the function and decide on your next action. You have 10 additional function invocations and 1 debugging check left.

**Assistant:** Based on the input-output pairs I have analyzed, I infer that this function is intended to evaluate whether all elements in a given list are distinct—that is, to determine if there are any duplicate entries within the list. To implement this functionality, I propose using a pairwise comparison strategy. By systematically comparing each element in the list with every subsequent element, it becomes possible to identify any duplicates that may exist. This approach ensures thorough coverage of all potential comparisons and provides a reliable method for detecting non-unique elements in the list.

**IMPLEMENTATION:**

```python
def solution(lst):
    # Use a simple pairwise comparison approach
    n = len(lst)
    for i in range(n):
        for j in range(i + 1, n):
            if lst[i] == lst[j]:
                return False
    return True
```

**Prompt:**

The implementation you generated in the previous iteration failed for the following input (and potentially other inputs):

```
Failed input: [1, 2, 3, 4, [5, 6], [5, 6]]
Ground Truth Function != Output From Generated Code:
'Error' != 'No Error'
```

This failure provides valuable information about a case you didn't account for. You should decide between generating more invocations to explore and debug the function behavior or generating one final implementation. You have 10 additional function invocations and NO debugging checks left.

**Assistant:** Based on the error message, it appears that raising an error for this input would be the appropriate response. To handle this effectively, a set-based approach might be more suitable than a pairwise comparison approach. By converting the list to a set, I can efficiently detect unhashable elements, as attempting this operation would naturally raise a TypeError for any such elements. Let me proceed with implementing this approach to validate the reasoning.

**IMPLEMENTATION:**

```python
def solution(lst):
    # Attempt to convert the list to a set.
    # If any element is unhashable, this will raise a TypeError as expected.
    return len(lst) == len(set(lst))
```

This final implementation passes the differential testing oracle.

## A.2 Supervised Fine-Tuning Experiment Setup

Table 8 presents the parameters used for our supervised fine-tuning experiments.

| Name | Value |
|---|---|
| Teacher Model | gpt-4o |
| Student Model | LLaMA-3.1-8B-Instruct |
| Fine-tuning Context Length | 32768 |
| Learning Rate | 1.00E-5 |
| LR Schedule | linear |
| Warmup Ratio | 0.03 |
| Epoch | 3 |
| Batch Size | 8 |
| Gradient Accumulation Steps | 8 |
| LoRA Rank | 8 |
| LoRA $\alpha$ | 8 |

Table 8: LoRA Fine-Tuning Parameters.

## A.3 Analysis of Differential Testing Tools

Program equivalence is undecidable, and no oracle can guarantee perfect correctness checking. To mitigate this, our evaluation combines two state-of-the-art differential testing tools, PYNGUIN (Lukasczyk & Fraser, 2022) and MOKAV (Etemadi et al., 2024). This dual-oracle setup reduces the likelihood of overlooking behavioral discrepancies between the synthesized and ground-truth functions.

Empirically, the two tools produce identical outcomes on 75.4% of problems. MOKAV detects additional bugs in 18.8% of cases missed by PYNGUIN, while the reverse holds in only 5.9%. Overall, using both tools yields a more reliable approximation of correctness in the absence of a perfect oracle.

