# OpenReview forum: "CodeARC: Benchmarking Reasoning Capabilities of LLM Agents for Inductive Program Synthesis"
_colmweb.org/COLM/2025/Conference — COLM 2025_

### Official Review · Reviewer_w6n7 · 2025-05-10

**Rating:** 7
**Confidence:** 5
**Ethics Flag:** 1

**Summary:**

This paper introduces CodeARC, a newly proposed benchmark aimed at evaluating the inductive reasoning abilities of large language models (LLMs) through the task of program synthesis from input-output examples. Unlike other current benchmarks based on static assessment, CodeARC introduces an interactive environment where models can ask a concealed target function and use a differential testing oracle to receive counterexamples. This supports iterative improvement of the produced code.
The benchmark consists of 1114 general-purpose Python functions, collected from existing datasets such as HumanEval, MBPP, and APPS. It includes both annotated and anonymized versions to test the model's reasoning capabilities, independent of function name cues. There are 18 LLMs tested, and the top-scoring model (o3-mini) scores just over 50% success, thus proving the difficulty of the task. The authors further demonstrate that fine-tuning using curated reasoning traces leads to a significant performance improvement, particularly for tasks involving meaningful identifiers.
Overall, CodeARC provides a useful and timely contribution to the state of the art, allowing better simulation of the extent to which LLMs can generalize from examples and learn from feedback, both essential abilities for the construction of intelligent autonomous agents.

**Questions To Authors:**

Task Diversity: Do you plan to expand the benchmark with more complex or multi-function programs that go beyond single-function synthesis? This might better evaluate multi-step reasoning or compositional abilities.
Oracle Quality: To what extent is the benchmark result sensitive to the quality or conduct of the differential testing tools (PYNGUIN, MOKAV)? Might some bugs or mismatches be overlooked or incorrectly reported?
Fine-Tuning Constraints: Did you observe any overfitting or performance degradation when fine-tuning on reasoning traces? How generalizable is the student model to unseen function structures?
Reinforcement Learning Potential: You mention RL as a future direction; do you have initial ideas for what a reward signal could be like in this synthesis context, or how you'd design such a system?

**Reasons To Accept:**

Novel Benchmark Design: The paper presents an interactive and realistic framework for inductive program synthesis that is ahead of static evaluation, addressing a major gap in existing LLM benchmarking.
Focus on Reasoning, Not Memorization: By using only input-output examples and using anonymized function names, CodeARC successfully isolates and tests genuine inductive reasoning instead of pattern-matching or retrieval.
Thorough Evaluation: The authors perform an extensive analysis of 18 models, including ablations on budget constraints and fine-tuning experiments, providing valuable insights into model behavior and limitations.
Practical Relevance: The reverse engineering–style configuration (querying a black-box function) reflects actual application scenarios like decompilation and debugging, making the benchmark extremely practical.

**Reasons To Reject:**

Limited Task Diversity: Although the benchmark covers 1,114 Python functions, many are drawn from existing datasets and mostly involve algorithmic problems. Adding more realistic, noisy, or domain-specific tasks would make the benchmark stronger and more generalizable.
Upper Bound on Difficulty: Even with the difficult setup, there are a lot of problems from HumanEval & MBPP  that still depend on fairly straightforward reasoning. The benchmark might be further reinforced by adding more multi-step or higher-abstraction synthesis problems that more effectively probe the limits of existing LLMs.

---

> ### Author Response · Authors · 2025-06-03
> **Response 1**
>
> We thank the reviewer for the constructive feedback.
>
> **Q1: Do you plan to expand the benchmark with more complex or multi-function programs that go beyond single-function synthesis? Adding more realistic, noisy, or domain-specific tasks would make the benchmark stronger and more generalizable**
>
>
> To partially address this concern, we include a preliminary evaluation on 200 Python functions randomly sampled from BigCodeBench [1], which are distinct from our original benchmark sources (HumanEval, MBPP, and APPS). This dataset reflects a broader range of real-world scenarios, including scientific computing (e.g., NumPy, SciPy, pandas), machine learning (e.g., scikit-learn), visualization (e.g., matplotlib), and systems programming. These problems often involve multiple function calls and require compositional reasoning to coordinate operations in a correct sequence.
>
> Model                    | HumanEval, MBPP, APPS | BigCodeBench
> -------------------------------------|---------------|----------------
> LLaMA-3.1-8B Base           | 19.3          | 32.5
> Fine-tuned LLaMA-3.1-8B | 25.3          | 40.0
> gpt-4o                                  | 37.8          | 46.5
> o1-mini                                | 53.2          | 40.0
> o3-mini                                | 59.5          | 43.0
>
> We observe that reasoning models (o3-mini, o1-mini) achieve lower success rates on BigCodeBench compared to their performance on the original benchmarks used in the paper (HumanEval, MBPP, and APPS). In contrast, non-reasoning models perform better on BigCodeBench. We attribute this to the nature of BigCodeBench tasks, which often involve domain-specific APIs and library usage, where non-reasoning models perform better.
>
> Finally, we emphasize that our evaluation protocol is modular and easy to extend. We have open-sourced our framework, and we encourage future researchers to build on it by adding more complex, multi-function, or domain-specific synthesis tasks.
>
> [1] Zhuo et al. BigCodeBench: Benchmarking Code Generation with Diverse Function Calls and Complex Instructions. ICLR 2025
>
>
> **Q2: To what extent is the benchmark result sensitive to the quality or conduct of the differential testing tools (PYNGUIN, MOKAV)?**
>
> We acknowledge that no oracle is perfect due to the undecidability of program equivalence. To mitigate this, we combine two state-of-the-art tools (Pynguin and Mokav) in our evaluation. Empirically, the tools agree in 75.4% of cases. Mokav detects additional bugs in 18.8% of cases missed by Pynguin, while the reverse holds in only 5.9%. Overall, using both tools yields a more reliable approximation of correctness in the absence of a perfect oracle.
>
>
> **Q3: Did you observe any overfitting or performance degradation when fine-tuning on reasoning traces? How generalizable is the student model to unseen function structures?**
>
> To mitigate overfitting, we carefully design the synthetic fine-tuning dataset to be disjoint from the evaluation set. All results reported in the paper are measured on our held-out benchmark.
>
> To explicitly assess overfitting, we evaluate model performance directly on the synthetic fine-tuning dataset:
>
>
> | Dataset     | Base Model (%) | Fine-Tuned (%) | Relative Improvement    |
> |-------------|----------------|----------------|-----------------|
> | Annotated   |    31.5       |      35.8     |     + 13.7%       |
> | Anonymized  |    11.4     |     15.4      |     +35.1%        |
>
>
> These relative improvements are comparable to those observed on the evaluation benchmark reported in Table 5 of the paper (+31% and +9.5%), suggesting that the fine-tuned model improves in a non-overfitted manner.
>
> To further evaluate generalization to structurally different and unseen functions, we conduct experiments on BigCodeBench (see Q1).
>
>
> Model                    | HumanEval, MBPP, APPS | BigCodeBench
> -------------------------------------|---------------|----------------
> LLaMA-3.1-8B Base           | 19.3          | 32.5
> Fine-tuned LLaMA-3.1-8B | 25.3          | 40.0
>
>
> Notably, our fine-tuned model shows a substantial gain over the base model on BigCodeBench (40.0% vs. 32.5%), indicating that the model fine-tuned on synthesis reasoning traces can *generalize to unseen datasets*.
>
>
> **Q4: You mention RL as a future direction; do you have initial ideas for what a reward signal could be like in this synthesis context, or how you'd design such a system?**
>
> For reward signal design, we consider several options. A straightforward approach is to assign positive reward only when the synthesized program passes the oracle, though this signal may be sparse. To provide denser feedback, partial rewards can be given for passing observed I/O examples, and penalties can be added for excessive use of I/O queries or oracle calls to promote efficiency. Our interactive evaluation setup in CodeARC naturally supports this design, making RL integration a promising direction for future work.

---

> > ### Comment · Reviewer_w6n7 · 2025-06-07
> >
> > Thanks for the response. I am happy with the response as it clarifies most of my concerns. I will keep the ratings the same.

---

> ### Author Response · Authors · 2025-06-06
> **Inquiry for post-rebuttal comments**
>
> Thank you once again for your helpful comments and suggestions. We have responded to each of your points with detailed explanations and additional results where appropriate. We would appreciate your thoughts on whether our responses have satisfactorily addressed your concerns. Please do not hesitate to share any further feedback or questions.

---

### Official Review · Reviewer_4Yd9 · 2025-05-12

**Rating:** 7
**Confidence:** 4
**Ethics Flag:** 1

**Summary:**

This paper proposes a new benchmark for measuring programming-by-example and inductive reasoning abilities in LLMs. The model is used as an agent that is initially provided with 10 I/O pairs for a function, and allowed to submit more inputs to obtain corresponding outputs, or submit its guess at the code for testing against an oracle, which can return an input for which the ground truth output doesn't match. The authors collect functions from the HumanEval+, MBPP+, and APPS datasets. The authors evaluate against 18 models and report the results.

**Reasons To Accept:**

The paper proposes a very interesting benchmark which meaningfully extends earlier ones focused purely on generating code in one shot from natural language descriptions and a few input/output examples.

The paper contains an extensive empirical study across a large number of models, and considers variations in the evaluation such as by varying the number of attempts that the model is given. The results show that the benchmark is far from saturated, and that expected trends are present (such as increasing performance with larger number of parameters, and a larger budget for oracle invocations).

**Reasons To Reject:**

Only one prompt/agent harness is tested in the paper, which may fail to elicit the maximal capabilities from the model. In particular, the model is expected to internally simulate the code that it generates, since it is not afforded with access to a Python interpreter. A better-equipped model, or one with different instructions, may exhibit significantly different performance than what the paper shows.

The authors specifically target more general-purpose tasks as opposed to ones used in prior work for programming-by-example; it would also have been good to see results on the classical programming-by-example tasks, since their usefulness is more thoroughly motivated by the body of prior work.

The use of up to 3 oracle invocations, as well as a budget of 30 input-output pairs, seems somewhat arbitrary. The paper motivates these choices as 'chosen based on practical constraints such as API cost and runtime". The difficulty of the dataset is highly contingent upon these parameters, and claims made in the paper may fail to hold up when they are increased. It's generally unclear whether we should expect all problems to be solvable given the budgets used, i.e. whether the programs are of sufficiently low complexity that it's possible to copy them exactly under these constraints.

There is currently no mention in the paper that the dataset will be released; the paper by itself, without the accompanying dataset/code, is not nearly as valuable.

---

> ### Author Response · Authors · 2025-06-03
> **Response 1**
>
> We thank the reviewer for the constructive feedback.
>
>
> **Q1: it would also have been good to see results on the classical programming-by-example tasks, since their usefulness is more thoroughly motivated by the body of prior work.**
>
>
> To complement our existing benchmark, we include preliminary results on 200 Python functions randomly sampled from BigCodeBench [1]. This dataset spans a wide range of real-world scenarios, including scientific computing (e.g., NumPy, SciPy, pandas), machine learning (e.g., sklearn), visualization (e.g., matplotlib), and systems programming, thereby expanding beyond the algorithmic focus of HumanEval, MBPP, and APPS.
>
> Model                    | HumanEval, MBPP, APPS | BigCodeBench
> -------------------------------------|---------------|----------------
> LLaMA-3.1-8B Base           | 19.3          | 32.5
> Fine-tuned LLaMA-3.1-8B | 25.3          | 40.0
> gpt-4o                                  | 37.8          | 46.5
> o1-mini                                | 53.2          | 40.0
> o3-mini                                | 59.5          | 43.0
>
> We observe that reasoning models (e.g., o3-mini) achieve lower success rates on BigCodeBench compared to their performance on the original benchmarks used in the paper (HumanEval, MBPP, and APPS). In contrast, non-reasoning models (e.g., gpt-4o) perform better on BigCodeBench. We attribute this to the nature of BigCodeBench tasks, which often involve domain-specific APIs and library usage, where non-reasoning models perform better.
>
> Notably, our fine-tuned model shows a substantial gain over the base model on BigCodeBench, indicating that the model fine-tuned on synthesis reasoning traces can *generalize to unseen datasets*.
>
> [1] Zhuo et al. BigCodeBench: Benchmarking Code Generation with Diverse Function Calls and Complex Instructions. ICLR 2025
>
>
>
> **Q2: There is currently no mention in the paper that the dataset will be released; the paper by itself, without the accompanying dataset/code, is not nearly as valuable.**
>
>
> We thank the reviewer for pointing this out. We have made both the dataset and code for CodeARC publicly available to support transparency and facilitate future research. For the purpose of double-blind review, we provide an anonymized link here:
>
>
> https://anonymous.4open.science/r/CodeARC-4808/README.md
>
>
>
> **Q3: The difficulty of the dataset is highly contingent upon the budget parameters, and claims made in the paper may fail to hold up when they are increased.**
>
> Our evaluation setup deliberately incorporates budget parameters to reflect real-world constraints on compute and time. Rather than being a limitation, this design choice makes the benchmark more realistic and practically grounded.
>
> Moreover, as shown in Table 2, models that achieve higher success rates tend to use fewer input-output queries and oracle invocations. This suggests that stronger models are able to solve problems more efficiently with fewer budgets. In this way, the budget not only controls the difficulty of the task but also serves as a signal of model efficiency and reasoning ability.
>
> Finally, we view the tunability of budget parameters as a key feature of CodeARC. As demonstrated by our ablation study, one can adjust the input-output and oracle budgets to scale the difficulty of the task. This stands in contrast to traditional static benchmarks and highlights CodeARC’s value as a flexible evaluation framework.
>
>
> **Q4: In particular, the model is expected to internally simulate the code that it generates, since it is not afforded access to a Python interpreter.**
>
>
> We acknowledge that our default evaluation setup restricts access to external tools such as a Python interpreter. However, our interactive protocol is flexible and can easily accommodate additional tools or actions. To assess this, we conducted a preliminary experiment by introducing an additional action that allows models to execute their generated code in a Python interpreter. We then updated the prompt instructions accordingly and re-ran the evaluation on 100 sampled problems from both the annotated and anonymized datasets. The results are shown below:
>
>
> | Model   | w/o Interpreter | w/ Interpreter |
> |---------|------------------|----------------|
> | gpt-4o  | 48.5             | 46.0           |
> | o3-mini | 69.0             | 71.0           |
>
>
> These results suggest that providing access to a Python interpreter does not uniformly improve model performance. For instance, while o3-mini benefits modestly (+2.0%), gpt-4o's performance slightly decreases (-1.5%). This outcome highlights that expanding the agent’s action space may sometimes lead to suboptimal behavior.
>
> Finally, we note that internal code simulation is a natural part of the inductive program synthesis process. Our benchmark is designed to evaluate inductive reasoning capability, and we believe that requiring models to reason without relying on external execution tools remains a meaningful and challenging setting.

---

> ### Author Response · Authors · 2025-06-06
> **Inquiry for post-rebuttal comments**
>
> Thank you again for your insightful comments and constructive suggestions. We have carefully addressed all the points you raised with detailed explanations and supporting experiments. We would be grateful to know if our responses have resolved your concerns. If you have any further questions or thoughts, we would be happy to continue the discussion.

---

> > ### Comment · Reviewer_4Yd9 · 2025-06-09
> >
> > Thank you for all your clarifications and additional experiments! I would like to maintain my score since it is already positive.
> >
> > With regards to BigCodeBench, it is nice to see the additional results but I don't know if it's really responsive to the request about "classical programming-by-example tasks". I believe that the addition of datasets seen in, for example, https://arxiv.org/abs/2406.08316 would improve the paper.

---

> > ### Author Response · Authors · 2025-06-11
> > **Response 2**
> >
> > Thank you again for your thoughtful follow-up and for maintaining a positive evaluation. We appreciate your suggestion to include classical programming by example (PBE) tasks, such as those discussed in [1], and would like to clarify our rationale for using BigCodeBench.
> >
> > Our main objective is to assess the inductive reasoning capabilities of LLMs in general-purpose programming languages such as Python. Many classical PBE datasets [1], such as FlashFill++ [2] and LambdaBeam [3], are defined over domain-specific languages with restricted syntax and specialized operators. While these datasets provide a valuable testbed, evaluating LLMs on them often **confounds inductive reasoning with the challenge of learning a new programming language with unfamiliar semantics**. In contrast, staying within Python allows us to **isolate and directly evaluate inductive reasoning, without introducing additional factors such as learning new syntax and semantics**.
> >
> > Furthermore, tasks in [1] focus on **three narrow domains**: list transformations, string editing, and turtle graphics. In comparison, our selected subset from **BigCodeBench spans a much broader set of real-world programming domains**, including numerical computing (NumPy and pandas), machine learning (scikit-learn), and data visualization (matplotlib). These domains require reasoning over widely used libraries and abstractions that are common in practical software development.
> >
> > Last but not least, the benchmark in [1] **is not currently released**. In contrast, **BigCodeBench is fully open source**.
> >
> > That said, we agree that evaluating inductive reasoning in domain-specific languages is an important direction. Our benchmark is meant to complement rather than replace such efforts. We hope our evaluation framework will inspire future research in this setting.
> >
> > **References:**
> >
> > [1] Li et al. Is Programming by Example Solved by LLMs? NeurIPS 2024.
> >
> >
> > [2] Cambronero et al. FlashFill++: Scaling Programming by Example by Cuing to the Chase. POPL 2023.
> >
> > [3] Shi et al. LambdaBeam: Neural Program Search with Higher-Order Functions and Lambdas. NeurIPS 2023.

---

### Official Review · Reviewer_3DHT · 2025-05-13

**Rating:** 7
**Confidence:** 3
**Ethics Flag:** 1

**Summary:**

This paper proposes a new evaluation framework for inductive program synthesis to evaluate the capability of LLMs in reasoning the underlying program from input-output examples. It provides an interactive environment where the LLM can request limited times for extra input-output examples as feedback for revision. This mitigates the ambiguity in inductive program synthesis that there may be multiple acceptable programs based on the given input-output pairs, and can evaluate LLMs' capability in function calls and self-correction based on feedback. The general-purpose inductive program synthesis benchmark, CodeARC, with 1114 functions, provides a realistic and challenging testbed.

**Questions To Authors:**

Section 5.3 shows that the increase of input-output query budget and the oracle invocation budget will consistently improve the performance. Is there an upper bound on the performance when it can no longer increase with more budget?
Among the query budget and the oracle invocation budget, which one is more important?

**Reasons To Accept:**

1. CodeARC provides an interactive test environment for inductive program synthesis, where the LLMs can request extra feedback by function calls. Unlike prior work, this can mitigate the problem of multiple valid functions satisfying the same input-output examples by providing additional feedback. It also evaluates the capability of the function call and revision based on feedback
2. It benchmarks 18 models on CodeARC and investigates the impact of budgets on input-output examples and function calls. The analysis on cases that pass the initial examples but fail under oracle testing indicates under-specification, which further motivates CodeARC.
3. It investigates how finetuning on the synthetic data affects the model performance on this benchmark

**Reasons To Reject:**

1. There is no ablation study on the input-output query and the oracle invocation. For example, if the IO budget is set to 0, will the oracle invocation benefit the performance? Similarly, if the oracle invocation budget is 0, how will the IO affect the performance? It is important to show that both of these mechanisms are important in CodeARC.
2. From Table 5, the improvement of the Success Rate on the Anonymized version is smaller than the Annotated version. This indicates that the finetuning on interactive reasoning traces on CodeARC cannot provide more benefits than a function name.

Missing Reference:
[1] Case2code: Learning inductive reasoning with synthetic data

---

> ### Author Response · Authors · 2025-06-03
> **Response 1**
>
> We thank the reviewer for the constructive feedback.
>
> **Q1: There is no ablation study on the input-output query and the oracle invocation. For example, if the IO budget is set to 0, will the oracle invocation benefit the performance? Similarly, if the oracle invocation budget is 0, how will the IO affect the performance?**
>
>
> In the original submission, we ablated the oracle and I/O budgets by varying one while keeping the other fixed at a non-zero default (Section 5.3). The reviewer proposes a complementary setup: fixing one budget to zero and varying the other to isolate its effect. We found this a valuable perspective and have conducted the additional experiments accordingly.
>
> The new results are shown below:
>
> ### Table 1: Effect of Oracle Invocations (with I/O Budget = 0)
>
> | Model     | Oracle = 0 | Oracle = 1 | Oracle = 2 |
> |-----------|------------|------------|------------|
> | o1-mini   |   33.7   |    43.7   |   44.1  |
> | o3-mini   |   35.9   |    51.3   |   54.2  |
>
>
>
> ### Table 2: Effect of Input-Output Queries (with Oracle Budget = 0)
>
> | Model     | I/O = 0 | I/O = 10 | I/O = 20 |
> |-----------|---------|----------|----------|
> | o1-mini   |  33.7  |     35.5    |   37.9   |
> | o3-mini   |  35.9  |     38.8    |   42.3   |
>
>
> When the I/O budget is zero, increasing the number of oracle invocations still substantially improves performance, as models benefit from counterexamples returned by the oracle. Conversely, even when the oracle budget is zero, additional I/O queries lead to consistent gains, as they provide more evidence of the target function's behavior.
>
> Together, these findings confirm the importance of both oracle budgets and I/O budgets.
>
>
>
>
> **Q2: Is there an upper bound on the performance when it can no longer increase with more budget? Among the query budget and the oracle invocation budget, which one is more important?**
>
>
>
> In real-world settings, agents operate under fixed budgets for both input-output queries and oracle invocations due to constraints on compute and time. While increasing either budget typically leads to better performance, the gains often diminish as the model saturates its capacity to refine hypotheses. For example, Figure 3 shows that gpt-4o-mini’s success rate plateaus with additional oracle invocations, and Table 4 indicates only a marginal improvement for o1-mini when increasing the I/O budget from 20 (49.6%) to 30 (50.5%). These observations reflect the diminishing returns of performance as budget allocations increase.
>
>
>
> While both budgets contribute to performance, oracle invocations appear more critical. As shown in our new ablation results (see Q1), increasing the oracle budget from 0 to 2 improves o3-mini’s success rate from 35.9% to 54.2%, while increasing the I/O budget from 0 to 20 improves performance from 35.9% to 42.3%. This suggests that oracle invocations provide more substantial improvements, likely because counterexamples from differential testing more directly identify and correct incorrect logic than additional I/O observations alone.
>
>
> **Q3: From Table 5, the improvement of the Success Rate on the Anonymized version is smaller than the Annotated version. This indicates that the finetuning on interactive reasoning traces on CodeARC cannot provide more benefits than a function name.**
>
>
> We agree with the observation that fine-tuning leads to a larger improvement on the annotated dataset compared to the anonymized version (31% vs. 9.5%). However, rather than suggesting that fine-tuning offers limited value, we interpret this result as highlighting the increased difficulty of the anonymized setting. Without any natural language cues such as function names, the anonymized version isolates the task of inductive reasoning and removes signals that might help shortcut the inductive reasoning process.
>
> As shown in Table 2, the anonymized dataset yields uniformly lower success rates across all models, indicating that it presents a fundamentally more challenging benchmark. Fine-tuning on curated synthesis traces still improves performance in this setting, but the gain is understandably smaller due to the absence of semantic hints and the more demanding nature of the task. We view this as a strength of the benchmark: it provides a rigorous testbed for evaluating true reasoning capabilities beyond name-based cues.
>
> Improving performance on anonymized synthesis remains an open challenge, and we believe future work, particularly using reinforcement learning or better-designed training traces, may help close this gap further.
>
>
>
> **Q4: Missing Reference: Case2code: Learning inductive reasoning with synthetic data**
>
> We thank the reviewer for pointing out this relevant work. We will cite it in our revised version.

---

> > ### Comment · Reviewer_3DHT · 2025-06-09
> >
> > I appreciate the authors' detailed response and additional experiments, which have addressed most of my concerns. I will maintain my current score, which is already sufficiently positive.

---

> ### Author Response · Authors · 2025-06-06
> **Inquiry for post-rebuttal comments**
>
> Thank you again for your thoughtful feedback and valuable comments. We have provided detailed responses and experimental results addressing all the questions you raised. We would greatly appreciate hearing from you on whether our rebuttal has sufficiently addressed your concerns. Please feel free to share any additional comments or follow-up questions.

---

### Official Review · Reviewer_G4M6 · 2025-05-14

**Rating:** 7
**Confidence:** 4
**Ethics Flag:** 1

**Summary:**

This paper proposes a new benchmark, CodeARC, for more general program by example tasks, with the target programs from existing coding benchmarks such as HumanEval, MBPP, and APPS.
It also introduces a new, agentic setup for inductive program synthesis, where the model is allowed to take actions, interact with a hidden oracle program, and decide when to submit its answer.
Several models are evaluated on CodeARC and the authors also show that fine-tuning on synthetic data (with the target program visible to the teacher model) can improve LLMs' performance on it.

**Reasons To Accept:**

1. Comprehensive evaluation and very thorough ablation experiments with interesting findings. For example, I find the results on whether initial I/O pairs are underspecified particularly interesting.

2. The agentic setup for evaluation is more realistic to how people may actually use LLMs for inductive program synthesis, and it opens doors to the study of scaling test-time compute for this task.

**Reasons To Reject:**

1. The "general-purpose" claim. The authors claim on Line 48-51 that "existing benchmarks ... are focused on domain-specific tasks and do not assess the ability of LLMs to synthesize functions written in general-purpose programming languages". I don't think it's true.
First, the tasks in CodeARC are not exactly general-purpose, they are mostly like interview questions and competition problems whose inputs and outputs are easily representable with strings. The tasks do not seem much more general-purpose than list operations to me.
Second, in Li and Ellis, 2025, a paper cited by the authors, LLMs are also tasked with solving PBE with a general-purpose programming language.

2. Lack of analysis on how fine-tuning affects models' ability on other coding tasks. Fine-tuning on synthetic data is great, but if it hurts other performance it wouldn't be worth it.


**Reference**

Wen-Ding Li and Kevin Ellis. Is programming by example solved by llms? Advances in Neural Information Processing Systems, 37:44761–44790, 2025

---

> ### Author Response · Authors · 2025-06-03
> **Response 1**
>
> We thank the reviewer for the constructive feedback.
>
> **Q1: The "general-purpose" claim**
>
> We appreciate the reviewer’s feedback and agree that our original phrasing slightly overstates the scope. In the revised version, we will tone down the claim and state that our benchmark makes a meaningful step toward a more general-purpose evaluation for inductive program synthesis, compared to existing datasets that are largely domain-specific.
>
> We also wish to clarify the distinction we make between *programming in a general-purpose language* and *benchmarks covering general-purpose tasks*. As the reviewer noted, prior benchmarks like Li and Ellis (2025) do use Python. However, they focus on three narrowly-scoped domains [1]:
>
>
> (1) List transformations that operate on lists of numbers;
>
>
> (2) Text editing tasks that perform string rewriting;
>
>
> (3) Logo-style turtle graphics with ASCII outputs.
>
>
> While written in a general-purpose language like Python, the underlying tasks remain domain-specific and algorithmically narrow.
>
> In contrast, our benchmark spans a broader range of programming logic and semantics. We start with programs drawn from HumanEval, MBPP, and APPS, which include more diverse algorithmic problems (e.g., numerical reasoning, data structure manipulation, simulation, etc).
>
> To further increase the coverage, we supplement our existing dataset with 200 Python functions randomly sampled from BigCodeBench [2], which includes problems involving scientific computing (e.g., NumPy, SciPy, pandas), machine learning libraries (e.g., sklearn), visualization (e.g., matplotlib), and system libraries. This addition enhances domain diversity beyond traditional algorithm-focused problems. We provide the following preliminary results to demonstrate model performance on this broader dataset.
>
> Model                    | HumanEval, MBPP, APPS | BigCodeBench
> -------------------------------------|---------------|----------------
> LLaMA-3.1-8B Base           | 19.3          | 32.5
> Fine-tuned LLaMA-3.1-8B | 25.3          | 40.0
> gpt-4o                                  | 37.8          | 46.5
> o1-mini                                | 53.2          | 40.0
> o3-mini                                | 59.5          | 43.0
>
> We observe that reasoning models (o3-mini, o1-mini) achieve lower success rates on BigCodeBench compared to their performance on the original benchmarks used in the paper (HumanEval, MBPP, and APPS). In contrast, non-reasoning models perform better on BigCodeBench. We attribute this to the nature of BigCodeBench tasks, which often involve domain-specific APIs and library usage, where non-reasoning models perform better.
>
>
> Finally, we emphasize that our interactive protocol is agnostic to the task domain and is easily extensible to any Python program. We will release the full benchmark and tooling upon publication, encouraging future work to scale up generality even further.
>
>
> [1] Wen-Ding Li and Kevin Ellis. Is Programming by Example Solved by LLMs? Advances in Neural Information Processing Systems 2025
>
>
> [2] Zhuo et al. BigCodeBench: Benchmarking Code Generation with Diverse Function Calls and Complex Instructions. ICLR 2025
>
>
> **Q2: Lack of analysis on how fine-tuning affects models' ability in other coding tasks**
>
> We thank the reviewer for raising this concern. To evaluate the impact of our fine-tuning on general code generation capabilities, we conduct a separate evaluation on BigCodeBench. The setup differs from our inductive program synthesis benchmark, as BigCodeBench evaluates traditional code generation from natural language descriptions involving diverse function calls and library usage. Furthermore, our synthetic fine-tuning dataset is constructed independently and contains no overlap with BigCodeBench problems. Thus, this serves as a clean test of the ability of the fine-tuned model on other coding tasks.
>
> We compare the pass@1 performance of the base and fine-tuned models on BigCodeBench:
>
>
> - Base model (meta-llama/Llama-3.1-8B-Instruct): 40.1%
>
>
> - Fine-tuned model: 39.6%
>
> We observe a marginal drop of 0.5 percentage points. This aligns with prior findings on **catastrophic forgetting**, a well-documented phenomenon where training on a new task can impair performance on previously learned ones [1]. While this small degradation is expected, we believe it is acceptable given the significant performance gains achieved on inductive program synthesis tasks. We will include this additional analysis and discussion in the final version.
>
> [1] Kotha et al. Understanding Catastrophic Forgetting in Language Models via Implicit Inference. ICLR 2024.

---

> > ### Comment · Reviewer_G4M6 · 2025-06-05
> >
> > Thank you for your clarification and the addition of BigCodeBench. I am now buying your claim about general-purpose and therefore raising my score to 7.

---

### Decision · Program_Chairs · 2025-07-08

**Decision:**

Accept

**Comment:**

This paper proposes an interesting twist to standard code benchmarks. Instead of being provided with the description and function signature, the model gets exposed only to input/output pairs and has to learn the underlying function. It does have the possibility to request feedback: either the output of a chosen input, or a counter-example to a proposed program

The authors benchmark 18 LLMs, showing poor performance (~50% max) on this task - even when the original prompts/completions were taken from benchmarks that are "solved" (performance >90%), and show the impact of finetuning. During the rebuttal phase the authors extended this to another benchmark that focuses more on software engineering and less on competitive programming

This is an excellent paper. The only point which is missing to me is the link to learnability theory: learning from positive examples and counter-examples has a rich history. The standard reference for that is Dana Angluin's "Learning Regular Sets from Queries and Counterexamples" from 1987, and several textbooks provide good context (eg, Colin de la Higuera's "Grammatical Inference: Learning Automata and Grammars")